# FlgV forms a flagellar motor ring that is required for optimal motility of *Helicobacter pylori*

Jack M. Botting[1☯], Shoichi Tachiyama[2,3☯], Katherine H. Gibson[1¤], Jun Liu[2,3], Vincent J. Starai[1], Timothy R. Hoover[1]*

1 Department of Microbiology, University of Georgia, Athens, Georgia, United States of America, 2 Microbial Sciences Institute, Yale University, West Haven, Connecticut, United States of America, 3 Department of Microbial Pathogenesis, Yale School of Medicine, New Haven, Connecticut, United States of America

☯ These authors contributed equally to this work.
¤ Current address: Centers for Disease Control and Prevention, Atlanta, Georgia, United States of America
* trhoover@uga.edu

## Abstract

Flagella-driven motility is essential for *Helicobacter pylori* to colonize the human stomach, where it causes a variety of diseases, including chronic gastritis, peptic ulcer disease, and gastric cancer. *H. pylori* has evolved a high-torque-generating flagellar motor that possesses several accessories not found in the archetypical *Escherichia coli* motor. FlgV was one of the first flagellar accessory proteins identified in *Campylobacter jejuni*, but its structure and function remain poorly understood. Here, we confirm that deletion of *flgV* in *H. pylori* B128 and a highly motile variant of *H. pylori* G27 (G27M) results in reduced motility in soft agar medium. Comparative analyses of *in-situ* flagellar motor structures of wild-type, Δ*flgV*, and a strain expressing FlgV-YFP showed that FlgV forms a ring-like structure closely associated with the junction of two highly conserved flagellar components: the MS and C rings. The results of our studies suggest that the FlgV ring has adapted specifically in Campylobacterota to support the assembly and efficient function of the high-torque-generating motors.

## Introduction

The human gastric pathogen *Helicobacter pylori*, a member of the phylum Campylobacterota, uses a cluster of polar sheathed flagella for motility, which is required for colonization in animal models for infection [1, 2]. The bacterial flagellum is organized into three basic parts, referred to as the basal body, hook, and filament [3–5]. The basal body contains a rotary motor powered by a transmembrane ion gradient and generates the torque needed for motility [6].

The torque-generating stator of the flagellar motor consists of a MotA pentamer and MotB dimer [7, 8]. MotB contains an N-terminal transmembrane domain, a plug/linker region, and a C-terminal periplasmic region that fixes the stator to the peptidoglycan layer through interactions with the peptidoglycan layer and P ring protein FlgI [9, 10]. MotA has four transmembrane helices and together with MotB forms a proton channel in the inner membrane [7, 8].

**Data Availability Statement:** Data has been deposited in the Electron Microscopy Data Bank

(https://www.ebi.ac.uk/emdb/). Entry code for the deposited data is EMD-40219.

**Funding:** This work was supported by National Institutes of Health grants AI140444 (TRH), AI1469077 (TRH), AI087946 (JL), and AI132828 (JI). The URL for NIH is https://www.nih.gov/. The funders had no role in study design, data collection and analysis, decision to publish, or preparation of the manuscript.

**Competing interests:** The authors have declared that no competing interests exist.

When not engaged with the motor, the MotB plug blocks proton flow through the stator [11]. Following incorporation of the stator into the motor, MotB undergoes a conformational change that unplugs the stator and promotes dimerization of the periplasmic domain to allow binding of the protein to peptidoglycan [12]. Proton flow through the stator is thought to power rotation of the MotA pentamer around the stationary MotB dimer [7, 8]. The C ring, which consists of the proteins FliG, FliM, and FliN (plus FliY in Campylobacterota) [13], forms the rotor of the flagellar motor [14, 15]. Torque from the stator units is transferred to the rotor through electrostatic interactions between conserved acidic and basic residues in MotA and the FliG Helix$_{Torque}$, an α-helix in the C-terminal domain of the protein [8]. The MS ring interfaces with the C ring through interactions between the C-terminal, cytoplasmic domain of FliF (FliF$_C$) and the N-terminal domain of FliG (FliG$_N$) [16, 17]. Engagement of MotA with the FliG Helix$_{Torque}$ results in rotation of the C ring/MS ring rotor complex, and the resulting torque is transmitted through the rod and hook to the filament, which acts as a propeller to push the cell forward [18, 19].

The MS ring is formed by the FliF protein and is one of the earliest flagellar structures assembled [20]. FliF is embedded in the inner membrane by two transmembrane helices that flank a large periplasmic region and contains the ring-building motif (RBM) domains RBM1, RMB2, RBM3a, and RBM3b, plus the β-collar domain [21, 22]. In addition to interacting with the C ring, the MS ring houses the flagellar type III secretion system (T3SS) that transports subunits of the axial components (e.g., rod, hook, and filament proteins) of the flagellum across the cell membrane [23]. Interestingly, FliF forms rings of mixed symmetry: an inner ring having 23-fold symmetry that interacts with the export gate of the T3SS and an outer ring with 34-fold symmetry that matches the symmetry of the C ring [24, 25].

*Campylobacter jejuni* is a member of the phylum Campylobacterota and is closely related to *H. pylori*. Gao and co-workers identified FlgV as a previously uncharacterized protein required for motility in *C. jejuni* and showed that the protein interacted with FliF in a co-immunoprecipitation assay [26]. However, the exact structure and function of FlgV remain to be defined. We report here that *H. pylori* and other members of the phylum Campylobacterota have FlgV homologs. Deleting *flgV* in *H. pylori* resulted in a reduction in the number of flagella per cell (i.e., hypoflagellation) and significantly impaired motility of the strain in soft agar medium. The hypoflagellation phenotype of the Δ*flgV* mutant did not fully account for the impaired motility of the strain, suggesting that FlgV plays roles in both flagellar assembly and function. Comparative analysis of *in-situ* structures of flagellar motors of *H. pylori* wild-type and Δ*flgV* mutants revealed that FlgV forms a ring closely associated with the MS ring and FliG$_N$. We hypothesize that the FlgV ring is an adaptation that allows the large-diameter, high-torque-generating motors of members of the phylum Campylobacterota to assemble and operate efficiently.

## Materials and methods

### Bacterial strains and growth conditions

*Escherichia coli* NEB TURBO cells were used for cloning and plasmid construction and grown in LB liquid or agar medium supplemented with ampicillin (100 μg/ml) or kanamycin (30 μg/ml) when appropriate. *H. pylori* strains and plasmids used in this study are listed in Table 1. *H. pylori* strains were grown microaerobically in an atmosphere consisting of 10% $CO_2$, 4–6% $O_2$, and 84–86% $N_2$ at 37°C on tryptic soy agar supplemented with 5% horse serum (TSA-HS) or Columbia agar supplemented with 5% sheep blood. Liquid cultures of *H. pylori* were grown in brain heart infusion (BHI) medium supplemented with 5% heat-inactivated horse serum with shaking in serum bottles in an atmosphere consisting of 5% $CO_2$, 10% $H_2$, 10% $O_2$, and 75%

N$_2$. *H. pylori* growth medium was supplemented with kanamycin (30 μg/ml) or sucrose (5%) when appropriate.

## Construction of *H. pylori* Δ*flgV* mutants

All primers used for PCR in the construction of *H. pylori* mutants are listed in S1 Table. Plasmids and *H. pylori* stains used in the study are listed in Table 1. Genomic DNA from *H. pylori* G27 was purified using the Wizard genomic DNA purification kit (Promega, Madison, WI, USA) and used as the PCR template to construct the suicide vectors used to generated deletion mutants. PCR was carried out using Phusion DNA polymerase (New England Biolabs, Ipswich, MA, USA), and the resulting amplicons were incubated with *Taq* polymerase (Promega) at 72˚C for 10 min to add 3'-A overhangs for T/A cloning with pGEM-T Easy plasmid (Promega). A 610 bp DNA fragment upstream of *flgV* was amplified using PCR primers 85 and 86, and a 600 bp downstream DNA fragment was amplified using PCR primers 87 and 88. PCR SOEing was used to join the upstream and downstream regions, and the resulting amplicon was cloned into pGEM-T Easy to generate plasmid pKHG25. Plasmid pKHG25 was cut at unique XhoI and NheI restriction sites and ligated with a kan$^R$-*sacB* cassette [27] to generate plasmid pKHG26. The suicide vector pKHG26 was introduced into *H. pylori* G27M or *H. pylori* B128 by natural transformation. Transformants in which the kan$^R$-*sacB* cassette on the plasmid had replaced the chromosomal copy of *flgV* by allelic exchange were selected on TSA-HS supplemented with kanamycin. Replacement of the kan$^R$-*sacB* cassette with *flgV* in kanamycin-resistant transformants was confirmed by PCR, and the resulting strains were

**Table 1. Plasmids and *H. pylori* strains used in this study.**

| Plasmid | Description | Source |
|---|---|---|
| pGEM-T Easy | TA cloning vector; Amp$^R$ | Promega |
| pKHG25 | pGEM-T Easy *flgV* flanking regions | This study |
| pKHG26 | pGEM-T Easy *flgV*::kan$^R$-*sacB* | This study |
| pKHG35 | pGEM-T Easy *flgV-yfp* | This study |
| pJC038 | pGEM-T Easy with kan$^R$-*sacB* cassette | [27] |
| pHel3 | *H. pylori* shuttle vector; Kan$^R$ | [28] |
| pHel3-GG | pHel3 containing tandem BspQ1 sites downstream of *fliF* promoter for Golden Gate cloning | This study |
| pJMB36 | pHel3-GG carrying *flgV* | This study |
| pJMB38 | pHel3-GG carrying *flgV* allele encoding FlgV$^{E71A,E72A}$ variant | This study |
| pJMB39 | pHel3-GG carrying *flgV* allele encoding FlgV$^{F15A,F16A}$ variant | This study |
| ***H. pylori* strain** | **Relevant genotype** | |
| G27 | wild type | D. Scott Merrell |
| G27M | *flgH178* (Gly178 to Cys) *fliL78* (Gln78 to stop) | This study |
| B128 | wild type | Richard M. Peek, Jr. |
| KHG45 | *H. pylori* G27M *flgV*::kan$^R$-*sacB* | This study |
| KHG48 | *H. pylori* G27M Δ*flgV* | This study |
| KHG58 | *H. pylori* G27M *flgV-yfp* | This study |
| KHG40 | *H. pylori* B128 *flgV*::kan$^R$-*sacB* | This study |
| KHG62 | *H. pylori* B128 Δ*flgV* | This study |
| JMB79 | *H. pylori* G27M Δ*flgV* / pJMB36 | This study |
| JMB81 | *H. pylori* G27M Δ*flgV* / pJMB38 | This study |
| JMB82 | *H. pylori* G27M Δ*flgV* / pJMB39 | This study |

designated KHG40 (B128 mutant) and KHG45 (G27M mutant). The suicide vector pKHG25 was introduced into strains KHG40 and KHG45 by natural transformation, and sucrose-resistant transformants in which the unmarked *flgV* deletion carried in plasmid pKHG25 had replaced the kan$^R$-*sacB* cassette in the chromosomal copy of *flgV* by allelic exchange were isolated by plating onto TSA-HS supplemented with 5% sucrose. Sucrose-resistant colonies were screened for sensitivity to kanamycin, and the deletion of *flgV* in a kanamycin-sensitive isolate was confirmed by PCR and DNA sequencing (Eton Bioscience, Research Triangle Park, NC, USA) of the resulting amplicon. The resulting G27M and B128 Δ*flgV* mutants were designated KHG48 and KHG62, respectively.

**Complementation of *flgV* mutation.** To facilitate complementation assays in *H. pylori*, we modified the shuttle vector pHel3 [28] for Golden Gate assembly using tandem sites for the Type IIS restriction enzyme BspQ1. The modified plasmid, which we designated pHel3-GG, contains the *fliF* promoter and the *ureA* ribosome-binding site upstream of the tandem BspQ1 sites where the cloned gene is introduced. Primers 151 and 152 were used to amplify *flgV* from *H. pylori* G27 genomic DNA. The primers introduced BspQ1 sites immediately upstream and downstream of the start and stop codons of *flgV*, respectively. The resulting amplicon and pHel3-GG were digested together with BspQ1 (New England Biolabs) for 1 h at 50˚C, after which the amplicon and vector were ligated using Fast-Link DNA ligase (Biosearch Technologies, Hoddesdon, UK) for 30 min at room temperature. The reaction mix was incubated with additional BspQ1 for 1 h at 50˚C, then used for transformation of *E. coli*. A plasmid containing the expected insert was identified following restriction enzyme digestion, and the identity of the insertion was confirmed by DNA sequencing (Eton Biosciences). The resulting plasmid, pJMB36, was introduced into KHG48 by natural transformation to generate strain JMB79.

**Site-directed mutagenesis of *flgV*.** Specific mutations were introduced into *flgV* carried on plasmid pJMB36 using the primer pairs 153 and 154, 155 and 156, or 157 and 158 together with the Q5 Site-Directed Mutagenesis Kit (New England Biolabs) per the manufacturer's instructions. The resulting plasmids, pJMB39 and pJMB38, were individually introduced into strain KHG48 by natural transformation to generate strains JMB81 and JMB82, respectively.

**Construction of *H. pylori* strain expressing FlgV-YFP fusion.** PCR primers 85 and 74 amplified *flgV* and an upstream flanking region totaling 877 bp, omitting the stop codon of *flgV*. Primers 75 and 76 were used to amplify *yfp* and added a sequence encoding a flexible linker (Gly-Ser-Ala-Gly-Ser-Gly) that joined *flgV* and *yfp*. Primers 77 and 78 amplified a 512 bp downstream sequence of *flgV*. PCR SOEing was used to join all three fragments, and the resulting amplicon was cloned into pGEM-T Easy to generate suicide vector pKHG35. Plasmid pKHG35 was introduced to strain KHG45 by natural transformation, and sucrose-resistant transformants in which the kan$^R$-*sacB* cassette in the chromosomal copy of *flgV* had been replaced with the *flgV-yfp* fusion by allelic exchange were isolated on TSA-HS supplements with kanamycin. The presence of the *flgV-yfp* fusion in the *flgV* locus was confirmed by PCR, and the strain was designated KHG58.

## Motility assay

Motility of *H. pylori* strains in a soft agar medium consisting of Mueller-Hinton broth, 10% heat-inactivated horse serum, 20 mM 2-(N-morpholino) ethanesulfonic acid (MES; pH 6.0), and 0.4% noble agar was tested as described [27]. Diameters of the swim halos emanating from the point of inoculation in the soft agar medium were measured 7 d post-inoculation. A minimum of three replicates were done for each strain. Mean values for swim halo diameters were calculated, and a two-sample *t* test was used to determine statistical significance.

## Transmission electron microscopy

*H. pylori* cultures were grown to late-log phase (A$_{600}$ = ∼ 1.0) in BHI supplemented with 10% heat-inactivated horse serum. Cells from the cultures were collected by centrifugation, fixed with formaldehyde and glutaraldehyde, and negatively stained with uranyl acetate as described [27]. Cells were visualized using a JEOL JEM 1011 transmission electron microscope operated at 80 kV.

## Genome sequencing and analysis

A genomic library of *H. pylori* G27M was prepared with the Illumina iTruSeq adaptor kit from 500 ng of gDNA and sequenced at the University of Georgia Genomics Facility by Illumina sequencing. Sequence analysis was performed using the *breseq* computational pipeline [29]. A reference genome was constructed by mapping reads on an NCBI genome for *H. pylori* G27 (Accession no.: NC_011333.1). Reads were mapped to the reference genome, and a minimum variant frequency of 0.8 was used to call variants and identify SNPs. Variants and SNPs in the strains were compared manually to identify those present only in the suppressor strains.

## Protein modeling and structure analysis

Tertiary structures of FlgV homologs and protein-protein interactions were predicted using the AlphaFold2 (ColabFold v1.5.1) tool in UCSF ChimeraX 1.5 [30]. Superimpositions of protein structures were performed using the Matchmaker tool in UCSF ChimeraX 1.5. All other visualizations and analyses of protein structures were conducted in UCSF ChimeraX 1.5.

## Cryo-electron tomography (cryo-ET)

*H. pylori* strains were grown on Columbia agar plates supplemented with 5% horse red blood cells at 37°C in 10% CO$_2$ conditions. For the complemented strains, the medium was supplemented with 30 μg/mL of kanamycin to maintain the shuttle vector. Bacteria from the agar medium were resuspended in phosphate buffer saline (PBS) and mixed with 10 nm of BSA gold tracers (Aurion, Wageningen, NL). The mixtures were deposited on freshly grow-discharged cryo-EM grids (Quantifoil R2/1, Cu 200, Ted Pella, Inc., Redding, CA, USA) and then frozen into liquid ethane using a manual plunger.

**Cryo-ET data collection and processing.** Frozen-hydrated specimens were visualized in a 300kV Titan Krios electron microscope (Thermo Fisher Scientific, Waltham, MA, USA) equipped with a K3 summit direct detection camera and a BioQuantum energy filter (Gatan, Pleasanton, CA, USA). Tilt series were acquired at 42,000x magnification (corresponding to a pixel size of 2.148 Å at the specimen level) using SerialEM [31] and FastTomo script [32] based on a dose-symmetric scheme at defocus ∼4.5 μm. The stage was tilted from -48° to +48° at 3° increments. The total accumulative dose for each tilt series was ∼60 e⁻/Å². MotionCor2 was used for drift correction [33]. Gold tracer beads were tracked to align all image stacks using IMOD [34]. Gctf was used for all aligned stacks to estimate defocus [35], and then the ctfphase-flip function in IMOD was used for contrast transfer function (CTF) correction.

**Subtomogram averaging.** Tomo3D [36] was used for 3D reconstruction, and a total of 268, 420, 224, 332, and 214 tomograms were reconstructed from the aligned tilt series of the wild-type, *ΔflgV* with *ΔfliL* (G27M), *ΔflgV* (B128), FlgV-YFP, and complemented strains, respectively. Binned 6 × 6 × 6 tomograms using the simultaneous iterative reconstructive technique (SIRT) were used to select 824, 2523, 654, 1255, and 659 flagellar motors at the bacterial pole from the wild-type, *ΔflgV* with *ΔfliL* (G27M), *ΔflgV* (B128), FlgV-YFP, and complemented strains, respectively. After particle picking, tomograms with weighted back projection

(WBP) were used for the initial subtomogram average structures using i3 suite [37, 38]. Binned $4 \times 4 \times 4$ subtomograms were used to refine the intact motor structures. Classification was used to remove non-flagellated motors.

## Results

### FlgV homologs are widespread among members of the phylum Campylobacterota

*H. pylori* 26695 was reported to lack a FlgV homolog [26], but this report was based on an incorrect annotation of the *flgV* locus that identified an open reading frame in the opposite orientation [39]. A subsequent analysis of the region revealed an open reading frame between *flhG* and *fliA* in the same orientation as these genes, and the gene was given the locus tag HP1033b (Fig 1A) [40]. The revised annotation of the *flhG-fliA* region was consistent with the genomes of other *H. pylori* strains, including *H. pylori* G27, where HPG27_395 is the HP1033b homolog. A blastp analysis revealed homology between HPG27_395 and *C. jejuni* FlgV, although the sequence similarity between the two proteins is low (30% identity/53% similarity over 89.4% of total length; Table 2). HPG27_395 and *C. jejuni* FlgV are of similar lengths (104 and 118 amino acid residues, respectively), and both are encoded in genes located between *flhG* and *fliA* (Fig 1B). FlhG is a MinD-like protein that controls flagellum number in many polar-flagellated bacteria [41, 42], while FliA ($\sigma^{28}$) is an alternative sigma factor needed for transcription of flagellar genes that are required late in the flagellum assembly pathway [43–45]. HPG27_395 and *C. jejuni* FlgV are predicted integral membrane proteins with similar membrane topologies (S1 Fig), and the tertiary structures of HPG27_395 and *C. jejuni* FlgV predicted by the AlphaFold2 tool in ChimeraX [30] are very similar (Fig 1C). Taken together, these data strongly suggest that HPG27_395 and *C. jejuni* FlgV are structurally and functionally equivalent, and hence we refer to HPG27_395 as FlgV.

A blastp search of the National Center for Biotechnology Information (NCBI) database revealed that homologs of *C. jejuni* FlgV are widespread among members of the phylum Campylobacterota (Table 2). In each case, the *flgV* homolog is located between *flhG* and *fliA* (Fig 1B). Although a blastp analysis failed to identify a FlgV homolog in *Arcobacter butzleri* 7h1h, the genome of the strain has a gene immediately downstream of *flhG* that encodes a protein similar in size, predicted membrane topology, and predicted tertiary structure to that of *C. jejuni* FlgV (Fig 1C). The failure to identify a FlgV homolog in *A. butzleri* from the blastp analysis was not unexpected as the *Arcobacter* flagellar motor proteins were reported to branch from the root of the Campylobacterota motor clade [47].

### FlgV is required for robust motility of *H. pylori* in soft agar medium

To determine if FlgV was required for flagellum function in *H. pylori*, we initially deleted *flgV* in *H. pylori* G27 and examined the motility of the resulting mutant in soft agar medium. Although the △*flgV* mutant produced a smaller swim halo than the G27 parental strain, the parental strain produced a relatively small swim halo, which limited our confidence in ascribing a motility defect to the △*flgV* mutant. To address this issue, we isolated a highly motile variant of *H. pylori* G27 from a motility enrichment and designated the motile variant as G27M. Whole genome sequencing of *H. pylori* G27M indicated that the strain has eighty single nucleotide polymorphisms (SNPs) and indels compared to an NCBI genome for *H. pylori* G27. The vast majority of the SNPs and indels presumably resulted from repeated passage of the G27 parental strain in the laboratory rather than during enrichment for enhanced motility. The SNPs and indels are in the coding regions of eight genes and twenty-nine pseudogenes (S2

Table), and twenty are in intergenic regions (S3 Table). G27M has intragenic mutations in two flagellar genes, *flgH* (encodes L ring protein) and *fliL* (encodes a motor protein). The mutation in *flgH* changed Gly-178 to Cys, while the mutation in *fliL* was a nonsense mutation that introduced a stop codon at codon 78 (Gln-78 to stop) (S2 Table). FliL is an integral membrane protein involved flagellar motor function in various bacteria [48–55]. *In-situ* structures of the *H. pylori* and *Borrelia burgdorferi* flagellar motors showed that the C-terminal periplasmic domain of FliL (FliL$_C$) forms a ring that surrounds the plug/linker regions of the MotB dimer in its extended, active form [53, 56].

Deleting *flgV* in *H. pylori* G27M resulted in hypoflagellation and a significant motility defect in soft agar medium (Fig 2A and 2B). Introducing *flgV* on the shuttle vector pHel3 into the G27M △*flgV* mutant complemented the motility and hypoflagellation defects, indicating that loss of FlgV was responsible for these phenotypes in the Δ*flgV* mutant.

Since G27M has mutations in *flgH* and *fliL*, we wished to determine if the phenotype of the G27M Δ*flgV* mutant is dependent on the mutations in these genes. To address this issue, we deleted *flgV* in *H. pylori* B128, which has wild-type *flgH* and *fliL* alleles, and characterized the resulting mutant. As observed with the G27M △*flgV* mutant, the B128 Δ*flgV* mutant was hypoflagellated and displayed a motility defect in soft agar medium (S2 Fig), indicating that the motility and flagellation defects in the G27M Δ*flgV* mutant were not reliant on the mutations in *flgH* or *fliL*.

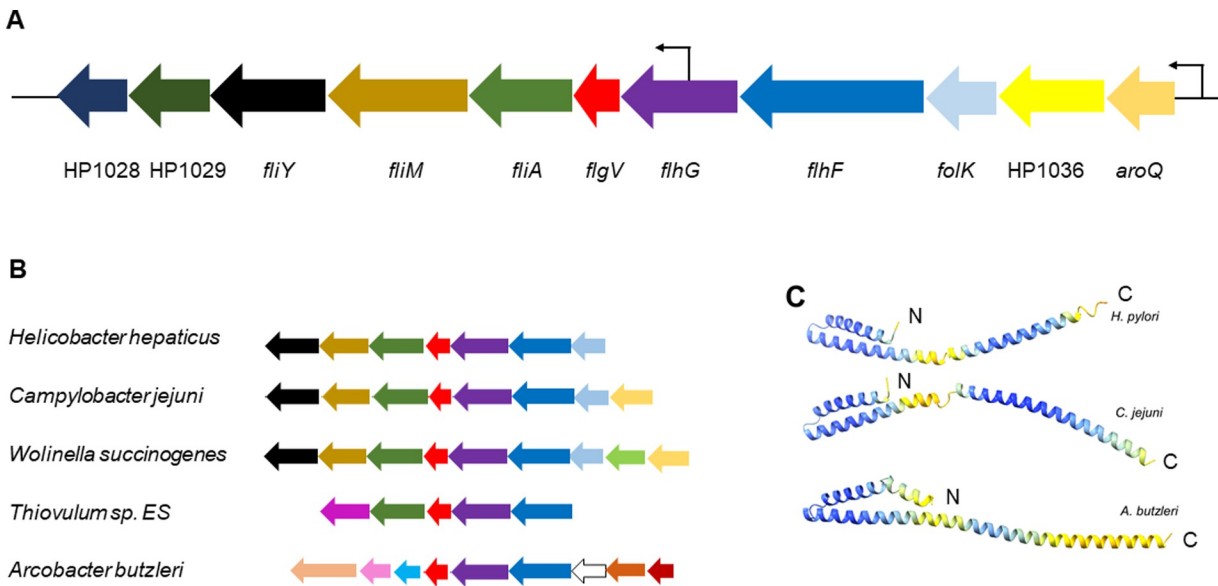

**Fig 1. Organization of operons containing *flgV* in *H. pylori* and representative members of the phylum Campylobacterota.** (**A**) Organization of genes within operon containing *flgV* (red) in *H. pylori* 26695. The smaller arrows upstream of *aroQ* and within *flhG* indicate identified transcriptional start sites in *H. pylori* 26695 [46]. Flagellar genes in the operon are *flhF* (dark blue), *flhG* (purple), *fliA* (green), *fliM* (dark gold), and *fliY* (black). Other genes are *aroQ* (gold), *folK* (light blue), and three genes of unknown function (HP1036, yellow; HP1029, dark olive; and HP1028, navy blue). (**B**) Synteny of *flhFGflgV* in representative genera of Campylobacterota. Homologous genes are color coded as indicated in panel A. Genes absent from the *H. pylori* 26695 *flhFGflgV* operon include *pepQ* (light green), potential *flgJ* homolog (dark red), potential *flgN* homolog (dark orange), hypothetical protein (white), *fliE* (light blue), *flgB* (pink), and *fliK* (peach). In *A. butzleri*, *flhFGflgV* appears to be part of a larger operon of ∼30 flagellar and chemotaxis genes. (**C**) Tertiary structures of FlgV proteins from *H. pylori*, *C. jejuni*, and *A. butzleri* predicted by the AlphaFold2 tool in ChimeraX [30]. The N-terminus (N) and C-terminus (C) of the protein are indicated. Regions in blue indicate a high confidence for predicted structure, while regions in yellow indicate a lower confidence for predicted structure.

**Table 2. _C. jejuni_ FlgV homologs in various genera of Campylobacterota identified by blastp analysis.**

| Description | Query coverage | Percent identity | Length | E value | Accession |
|---|---|---|---|---|---|
| hypothetical protein [_Campylobacter jejuni_ 81–176] | 100% | 100.00% | 118 | 5e-80 | WP_002868772.1 |
| hypothetical protein [_Helicobacter pylori_ G27] | 78% | 30.11% | 104 | 1e-07 | WP_000868000.1 |
| hypothetical protein [_Helicobacter hepaticus_ ATCC 51449] | 43% | 43.14% | 118 | 1e-09 | WP_011115985.1 |
| hypothetical protein [_Wolinella succinogenes_ DSM 1740] | 94% | 27.59% | 117 | 2e-09 | WP_011139455.1 |
| hypothetical protein [_Sulfurimonas autotrophica_ DSM 16294] | 86% | 38.24% | 111 | 1e-05 | WP_013326520.1 |
| hypothetical protein [_Sulfurospirillum multivorans_ DSM 12446] | 91% | 41.05% | 124 | 5e-15 | WP_025343576.1 |
| hypothetical protein [_Sulfuricurvum kujiense_ DSM 16994] | 75% | 30.34% | 107 | 3e-11 | WP_013459341.1 |
| hypothetical protein [_Hydrogenimonas thermophila_] | 75% | 33.71% | 109 | 1e-09 | WP_092910690.1 |
| hypothetical protein ThvES_00009150 [_Thiovulum_ sp. ES] | 84% | 28.95% | 112 | 4e-04 | EJF07001.1 |
| hypothetical protein [_Caminibacter mediatlanticus_ TB-2] | 77% | 40.82% | 112 | 7e-08 | WP_007473940.1 |
| hypothetical protein [_Lebetimomas natshushimae_] | 77% | 39.13% | 112 | 2e-11 | WP_096259429.1 |
| hypothetical protein [_Nautilia profundicola_ Am-H] | 77% | 36.17% | 111 | 2e-09 | WP_012663466.1 |
| hypothetical protein [_Arcobacter butzleri_ 7h1h] | [a]nssf | [a]nssf | 127 | >5e-03 | WP_014469277.1 |

[a]No significant similarity found

## Conserved phenylalanine residues in the _H. pylori_ FlgV transmembrane region are required for flagellum assembly

Alignment of the _H. pylori_ G27 FlgV sequence with the sequences of FlgV homologs from nine representative species of Campylobacterota genera revealed few conserved amino acid residues (S3 Fig). A notable exception, however, was a GFFxG motif in the first transmembrane helix of _H. pylori_ FlgV that was present in all the FlgV homologs we examined. GxxxG motifs in transmembrane regions are often involved in protein-protein interactions [57]. The Phe residues in the GFFxG motif of _H. pylori_ FlgV (Phe-15 and Phe16) were changed to Ala residues to examine the potential role of these conserved residues in FlgV function. In addition to introducing alanine substitutions in the conserved Phe-15 and Phe-16 residues, the conserved residue Glu-71 together with adjacent amino acid residue Glu-72 were also changed to Ala residues. Glu-71 and Glu-72 are located in the C-terminal domain of FlgV (FlgV$_C$).

Both the FlgV$^{F15A,F16A}$ and FlgV$^{E71A,E72A}$ variants supported robust motility of the G27M Δ_flgV_ mutant in soft agar medium (Figs 2A and S4). The strain expressing the FlgV$^{E71A,E72A}$ variant was well flagellated and had significantly more flagella per cell compared to the G27M Δ_flgV_ parental strain (Fig 2B). By contrast, the strain expressing the FlgV$^{F15A,F16A}$ variant was hypoflagellated (Fig 2B), and about half the cells examined lacked flagella. The FlgV$^{F15A,F16A}$ variant failed to suppress the flagellation defect of the Δ_flgV_ mutant but did rescue motility in the Δ_flgV_ mutant, indicating that the hypoflagellation of the Δ_flgV_ mutant does not account fully for its motility defect. Thus, FlgV plays apparent roles in both flagellar assembly and function.

Since FliF is a FlgV interaction partner [26], we modeled potential interactions between FlgV and FliF using AlphaFold2. The FlgV-FliF interaction predicted with the highest confidence involved the transmembrane helices of the proteins (Fig 2C). Interactions between the transmembrane helices involved contact at several points, including an interface between the conserved Phe-15 and Phe-16 in FlgV and Phe-466 and Tyr-467 in FliF, which are located in the transmembrane helix near the C-terminus (Fig 2D). At this position, the chains were predicted to be within 4.4 Å of each other, and each of the four residues was a predicted contact point, with at least 15 Å$^2$ of buried solvent-accessible surface area. Modeling interactions of the FlgV$^{F15A,F16A}$ variant with FliF revealed that Phe-466 and Tyr-467 in FliF and Ala-16 in FlgV

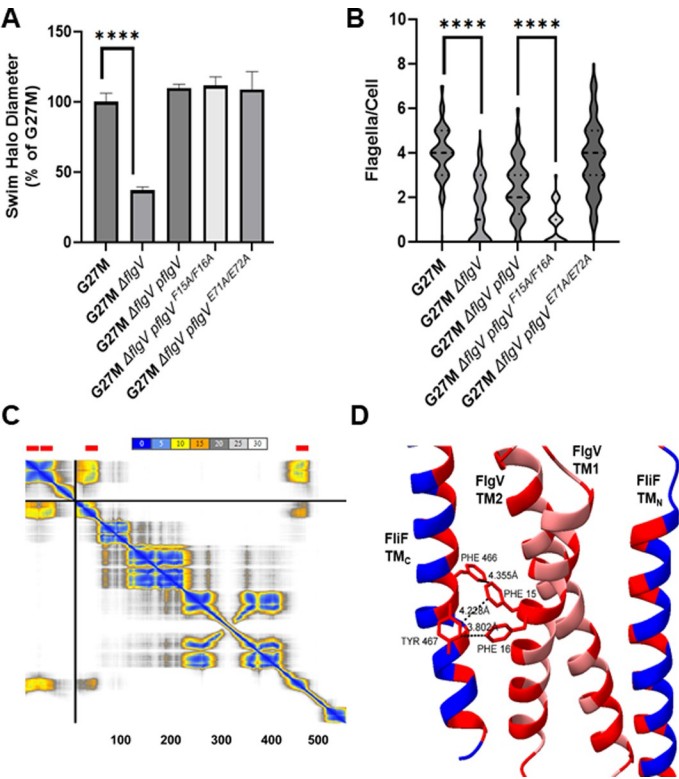

**Fig 2. Motility and flagellation phenotypes of the *H. pylori* G27M Δ*flgV* mutant.** (**A**) Motility of *H. pylori* G27M, Δ*flgV* mutant (G27M Δ*flgV*), and Δ*flgV* mutant complemented with plasmid-borne copies of wild-type *flgV* (G27M Δ*flgV* p*flgV*), *flgV* allele encoding FlgV[F15A,F16A] variant (G27M Δ*flgV* p*flgV*[F15A/F16A]), or *flgV* allele encoding FlgV[E71A, E72A] variant (G27M Δ*flgV* p*flgV*[E71A,E72A]). Strains were stab inoculated into soft agar medium, and diameters of the resulting swim halos were measured following 7 d incubation. Mean values for swim halo diameters of each strain were normalized to the mean swim halo value for *H. pylori* G27M. Number of technical replicates for the strains ranged from 3 to 12. Error bars indicate the standard deviation of the mean. Asterisks indicate swim halo diameters that differed significantly from those of G27M ($p$-value <0.00001). Statistical analysis of the data was done using a two-sample *t* test. (**B**) Flagella were counted for at least 100 cells per strain. Asterisks indicate numbers of flagella per cell that were significantly different ($p$-value <0.00001) as determined using a Mann-Whitney U test. (**C**) Prediction aligned error (PAE) score for the model for interactions between FlgV and FliF. Both the x- and y-axis indicate the position of individual amino acids. The numbers on the x-axis indicate the approximate amino acid positions for FliF. The uncertainty in the predicted distance of two amino acids is color coded from blue (0 Å) to white (30 Å), as indicated by the bar at the top of the figure. The upper left quadrant corresponds to errors in the distances of residues within FlgV, and the lower right quadrant to errors within FliF. Transmembrane helices in FlgV and FliF are indicated by the red bars at the top of the figure. Note the lower error for the distances of the transmembrane helices of the two proteins as seen in the lower left and upper right quadrants. (**D**) AlphaFold2 predictions of protein-protein interactions between transmembrane helices of FliF (blue) and FlgV (pink) modeled in ChimeraX [30]. Regions highlighted in red are predicted points of contact between the proteins. Predicted distances between residues Phe-15 and Phe-16 in FlgV and residues Phe-466 and Tyr-467 in FliF are indicated. FliF C-terminal transmembrane helix (FliF TM_C), FliF N-terminal transmembrane helix (FliF TM_N), first transmembrane helix in FlgV (FlgV TM1), second transmembrane helix in FlgV (FlgV TM2).

no longer served as contact points, and the distances between the chains increased from 4.4 Å to between 5.8 and 8.9 Å. Alignment of the predicted C-terminal transmembrane helices of FliF homologs from representatives of various genera of Campylobacterota indicated that *H. pylori* FliF residues Phe-466 and Tyr-467 are well conserved across diverse Campylobacterota genera (S5 Fig).

### FlgV forms a ring-like structure near the junction of the MS and C rings

The motility defects of the *H. pylori* G27M and B128 △*flgV* mutants suggested that the mutant motor lacks a component critical for optimum motility. To address this hypothesis, we compared *in-situ* structures of the motors of wild-type *H. pylori* G27, *H. pylori* G27M Δ*flgV* mutant, *H. pylori* B128 Δ*flgV* mutant, and *H. pylori* G27M Δ*flgV* mutant complemented with *flgV*. *In-situ* structures of the motors were generated by cryo-ET and subtomogram averaging. The motor of wild-type *H. pylori* G27 (Fig 3A) has a globular density on the cytoplasmic side of the inner membrane near the junction of the C ring and MS ring that is absent from the motors of the G27M Δ*flgV* and B128 Δ*flgV* mutants (Fig 3B and 3C). A corresponding density at the junction of the C ring and MS ring was noted previously for the *C. jejuni* flagellar motor [58]. The globular density corresponds to a ring-like structure (Fig 3E) absent from the motors of the B128 Δ*flgV* and G27M Δ*flgV* mutants (Fig 3F and 3G). These data suggest that the previously uncharacterized *H. pylori* motor ring is formed by FlgV.

To further assess whether FlgV does indeed form the novel motor ring, we expressed a FlgV-yellow fluorescent protein (YFP) fusion from the native *flgV* locus in *H. pylori* G27M. The motility of the strain expressing the FlgV-YFP fusion protein in soft agar medium did not differ from that of G27M (S6 Fig). The motor structure of the G27M strain expressing the FlgV-YFP fusion protein showed that the globular density near the C ring is present (Fig 3D), as well as the ring structure internal to the C ring (Fig 3H). An additional globular density was associated with the motor of the G27M strain expressing the FlgV-YFP fusion protein (Fig 3D), which corresponded to YFP. These data provide compelling evidence that FlgV forms the motor ring structure internal to the C ring, which we hereafter refer to as the FlgV ring. Consistent with the nonsense mutation in *fliL* identified from the whole genome sequencing of G27M (S2 Table), motors of strains derived from G27M lacked the $FliL_C$ ring (Fig 3K and 3L).

Closer examination of the FlgV ring showed that the cytoplasmic portion of the protein forms a ring with 21-fold symmetry that is approximately 33 nm in diameter (Fig 4C). The cytoplasmic portion of FliF, $FliG_N$, and $FlgV_C$ are also closely associated and located within the same plane (Fig 4C and 4E, and 4F). Symmetries of the FlgV ring and $FliG_N$ (and C ring) differ though, as the latter has 39-fold symmetry (Fig 4C and 4F).

### Discussion

The *H. pylori* flagellar motor, as well as the motors of many other bacteria, have accessories not found in *E. coli* and *S. enterica* motors [47, 58–61]. Identifying and characterizing the proteins that form these motor accessories is essential for revealing the unique adaptations that have evolved to support specific flagellar functions in different bacterial species. We report here a previously uncharacterized *H. pylori* motor accessory formed by FlgV. Consistent with the previous report that *C. jejuni* FlgV interacts with FliF [26], this FlgV ring is intimately associated with the MS ring in the *H. pylori* motor (Figs 3 and 4). More importantly, we found that the FlgV ring is located at the junction between the MS and C rings. Given that the MS ring and C ring are highly conserved and essential for flagellar assembly and function, the FlgV ring likely plays an important role through interactions with key components of the motor. FlgV homologs are found in many members of the phylum Campylobacterota (Table 2), and we expect the motors of these bacteria to contain the FlgV ring.

The synteny of *flhFG* and *flgV* within Campylobacterota genomes is notable as it is maintained in diverse genomic contexts (Fig 1B). In many Campylobacterota species, including *H. pylori*, *flhFGflgV* is immediately upstream of the flagellar genes *fliAMY*. As in *H. pylori*, the *flhFGflgV* locus in the *A. butzleri* genome is part of a larger flagellar operon, but the surrounding flagellar genes differ from those in the genomes of *H. pylori* and other Campylobacterota (Fig 1B). FlhF is

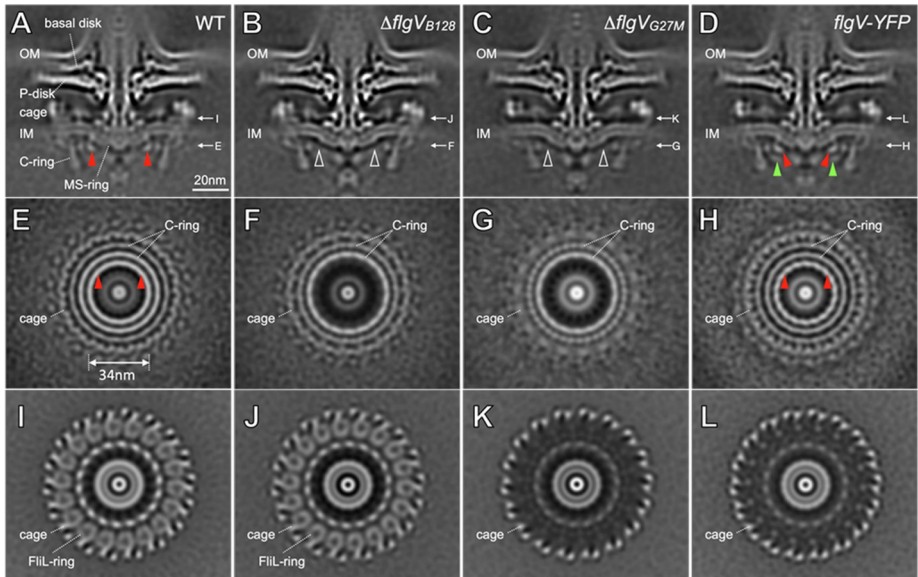

**Fig 3. *In-situ* structures of the flagellar motor in *H. pylori* G27 wild-type and *flgV* mutant cells.** (**A-D**) Medial slices through *in-situ* structures of G27 WT and several *flgV* mutant motors determined by cryo-ET and subtomogram averaging, respectively. A ring-like structure (highlighted by red arrows) in the G27 WT motor (**A**) is absent as shown by white arrows in the B128 Δ*flgV* mutant (Δ*flgV*$_{B128}$) (**B**) and G27M Δ*flgV* mutant (Δ*flgV*$_{G27M}$) (**C**). (**D**) Extra densities highlighted by green arrows are adjacent to the ring-like structure (red arrows) in G27M expressing FlgV-YFP. (**E-H**) Perpendicular cross sections at the top of the C ring through the G27 WT and *flgV* mutant motor structures corresponding to the panels (**A-D**), respectively. Note the absence of electron densities corresponding to the FlgV ring from two Δ*flgV* mutants (**F, G**). (**I-L**) Different perpendicular cross sections at the FliL ring level through the WT and *flgV* mutant motor structures corresponding to the panels (**A-D**), respectively. Note the absence of the FliL ring from the motors of the two mutants derived from strain G27M (**K, L**). OM, outer membrane; IM, inner membrane.

a GTPase belonging to the signal recognition particle family that includes Ffh and FtsY, which direct protein substrates to the Sec system for secretion or insertion in the cell membrane [62]. FlhF cycles between a GTP-bound form that facilitates assembly of the MS ring at the cell pole in many polar-flagellated bacteria, and an inactive GDP-bound or apo-form [42]. In polar-flagellated bacteria, mutations in *flhF* typically result in hypoflagellation and the mislocalization of flagella to non-polar sites [63–66]. FlhG controls flagella number by stimulating the GTPase activity of FlhF, and mutations in *flhG* often result in hyperflagellation [67–71].

As in other bacteria, deletion of *flhF* in *H. pylori* results in hypoflagellation and the mislocalization of flagella to non-polar sites. Given that the Δ*flgV* mutants in both the G27M and B128 backgrounds were hypoflagellated (Figs 2B and S2B), we hypothesize that FlgV works in conjunction with FlhF to facilitate assembly of the MS ring in *H. pylori*. We did not observe the mislocalization of flagella to non-polar sites in the △*flgV* mutants though, which suggests that FlgV is not required for FlhF activity.

An intriguing question is why might FlgV be needed for MS ring assembly in *H. pylori* but not in *S. enterica*? The answer may lie in the complicated assembly of the MS ring, which involves the complex arrangement of subunits and mixed symmetries built from a single polypeptide chain [24]. While the symmetries of the *H. pylori* FliF rings have not been defined, it is safe to assume that the outer ring has a 39-fold symmetry that matches the symmetry of the C ring (Fig 4C and 4F) given that the *S. enterica* outer ring matches the 34-fold symmetry of its C ring. We expect that the *H. pylori* FliF inner ring has the same 23-fold symmetry found in the *S. enterica* FliF inner ring since it houses the export gate of the flagellar T3SS, which is highly

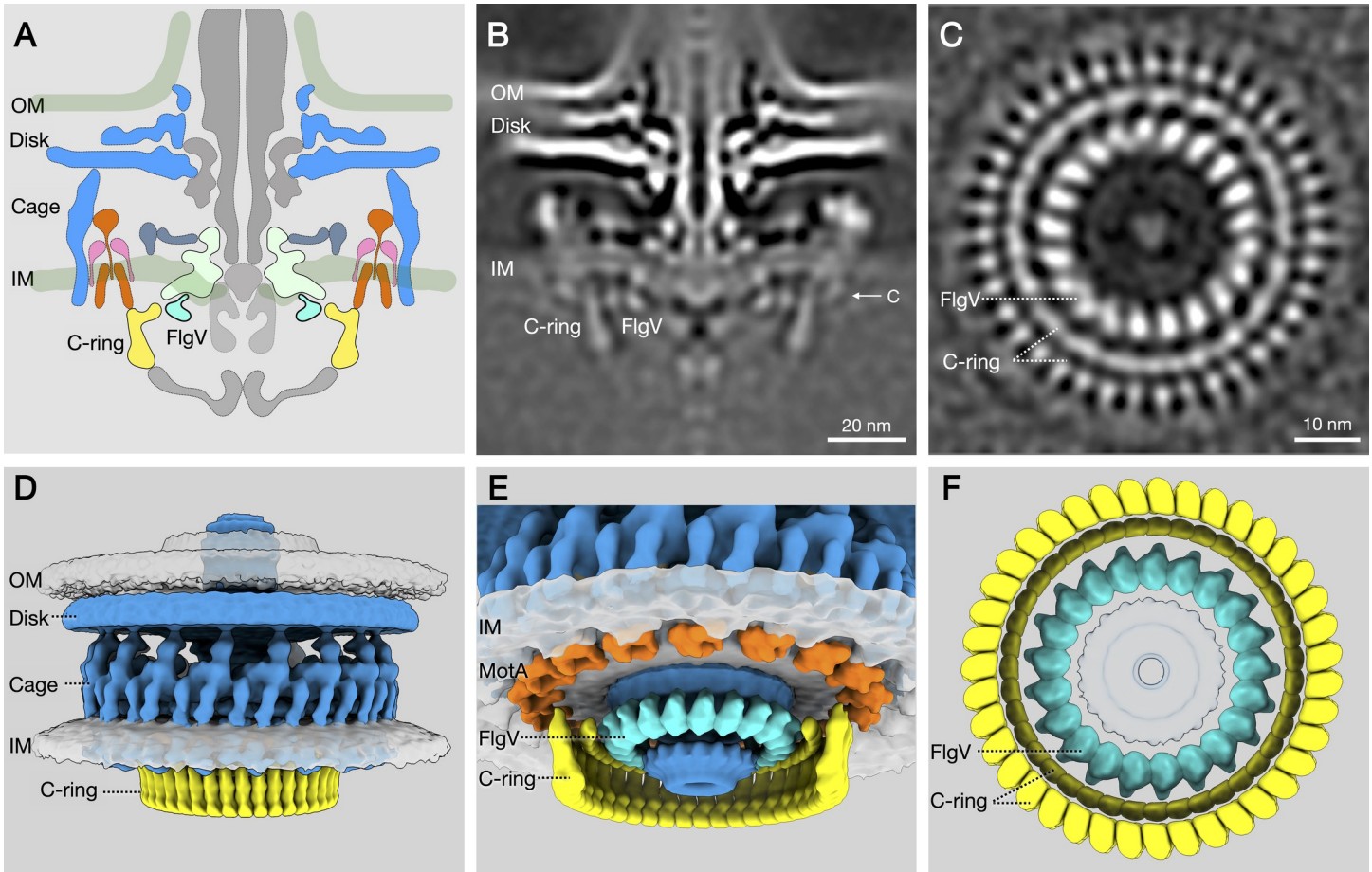

**Fig 4. *In-situ* structure of the *H. pylori* FlgV ring.** (**A**) Cartoon of the *H. pylori* flagellar motor. The FlgV ring (cyan) is located near the junction of the C ring (yellow) and MS ring (light green). Core motor components (flagellar protein export apparatus, rod, hook, P ring, and L ring) are shown in gray. Stator proteins MotA and MotB are shown in orange. FliL is shown in pink. *H. pylori* motor accessories are shown in dark blue and include the cage-like structure, outer disk, and basal disk. Outer membrane (OM); inner membrane (IM). (**B**) Medial slice through subtomogram averaged structure of the *H. pylori* G27 flagellar motor. (**C**) Perpendicular cross section at the top of the C ring through the FlgV ring. Note the 39-fold symmetry of the C ring and 21-fold symmetry of the FlgV ring. (**D-F**) Isosurface renderings of *H. pylori* motor from different vantage points. (**D**) Exterior side view of the motor showing the basal disk, cage, and C ring. (**E**) View of motor tilted to show structures exposed on the cytoplasmic side of the inner membrane and location of FlgV ring (cyan) relative to MS ring (dark blue), C ring and FliG$_N$ (both in yellow), and MotA pentamers of the stator units (orange). (**F**) Perpendicular cross-sectional view of motor from the cytoplasmic side of the inner membrane.

conserved across different bacterial species. The greater number of FliF subunits in the *H. pylori* MS ring compared to the *S. enterica* MS ring may impose barriers in assembling an inner ring with a 23-fold symmetry. FlgV may overcome these barriers and facilitate building of the FliF rings. For example, interactions between the transmembrane helices of specific FlgV and FliF monomers may confine the movement of these FliF subunits and promote assembly of the inner ring from their RBM2s. In the absence of FlgV, FliF rings with incorrect symmetries may be formed that interfere with assembly of the T3SS export gate or adversely affect interactions between FliF$_C$ and FliG$_N$. Such outcomes could result in flagellum assembly being aborted or might mark the flagellum for disassembly.

In addition to its apparent role in MS ring assembly, FlgV plays a role in motor function, as supported by the observation that the FlgV$^{F15A,F16A}$ variant failed to suppress the flagellation defect of the G27M Δ*flgV* mutant but did support robust motility of the mutant (Fig 2A and 2B). These findings indicate that the hypoflagellation of the G27M Δ*flgV* mutant does not account for its decreased motility as the strain expressing the FlgV$^{F15A,F16A}$ variant displayed a

similar degree of hypoflagellation but was highly motile. The MS ring not only houses the T3SS export gate but also interfaces with the C ring to form the rotor. The connection between the MS ring and C ring involves $FliG_N$ folding around two helices of $FliF_C$ near its C-terminus [17]. The MS ring-C ring junction is critical for flagellar function, and FlgV may enhance the motility of *H. pylori* in soft agar medium by stabilizing this junction. Stabilizing the MS ring-C ring junction may hold particular importance for bacteria that have high-torque-generating motors, such as *H. pylori* and other Campylobacterota. Another possible mechanism by which the FlgV ring optimizes motor function is by facilitating productive interactions between the FliG $Helix_{Torque}$ and MotA. There is precedence for auxiliary proteins modulating interactions between the stator and rotor, as the *Bacillus subtilis* motor clutch proteins EpsE and MotI act to disengage the stator units from the rotor [72, 73].

Alternatively, the FlgV ring may assist the switch complex in controlling the rotation direction of the rotor. In the bacterial chemotaxis system, CheY-phosphate induces a directional switch that causes the FliG $Helix_{Torque}$ to engage the opposite side of the MotA pentamer [8, 74, 75], resulting in the MotA pentamer powering rotation of the rotor in the clockwise direction. It is possible that the FlgV ring aids in motor reversals important for agar-based motility. Examining the swimming behavior of the *H. pylori* △*flgV* mutants and △*flgV* motile variants should allow us to distinguish between these models for FlgV function.

## Supporting information

**S1 Fig. Predicted membrane topology of *C. jejuni* and *H. pylori* FlgV homologs.** The amino acid sequences of *C. jejuni* 81–176 and *H. pylori* G27 FlgV homologs were analyzed using DeepTMHMM (https://dtu.biolib.com/DeepTMHMM) to predict transmembrane topology of the proteins. Predicted transmembrane helices are indicated in orange, regions of the protein predicted to be exposed on the cytoplasmic side of the membrane are indicated by the pink line, and regions of the protein predicted to be exposed on the periplasmic side of the membrane are indicated by the blue line.
(TIF)

**S2 Fig. Motility and flagellation phenotypes of *H. pylori* B128 Δ*flgV* mutant.** (**A**) Motilities of *H. pylori* B128 wild type and *H. pylori* B128 Δ*flgV* mutant in soft agar medium. Strains were stab inoculated into soft agar medium, and diameters of the resulting swim halos were measured following 7 d incubation. Bars indicate mean values for swim halo diameters. Three replicates were done for each strain. Error bars indicate standard deviation of the mean. The swim halo diameter of the Δ*flgV* mutant differed significantly from that of wild type (*p*-value <0.00001). Statistical analysis of the data was done using a two-sample *t* test. (**B**) Flagella were counted for at least 95 cells for *H. pylori* B128 wild type and *H. pylori* B128 Δ*flgV*. Distribution of the number of flagella per cell for Δ*flgV* mutant differed significantly from that of wild type (*p*-values <0.00001). Statistical significance for differences in the distribution of the number of flagella per cell were determined using a Mann-Whitney U test.
(TIF)

**S3 Fig. Alignment of amino acid sequences of FlgV homologs.** Amino acid sequences of FlgV homologs from *H. pylori* G27 (Hpy), *Wolinella succinogenes* DSM 1740 (Wsu), *Campylobacter jejuni* 81–176 (Cje), *Lebetimonas natshushimae* (Lna), *Caminibacter mediatlanticus* (Cme), *Nautilia profundicola* AM-H (Npr), *Sulfurimonas autotrophica* DSM 16294 (Sau), and *Hydrogenimonas thermophila* (Hth) were aligned using Clustal Omega (https://www.ebi.ac.uk/Tools/msa/clustalo/). Transmembrane helices 1 and 2 (TM-1 and TM-2) for *H. pylori* FlgV are indicated and were predicted using Phobius (https://www.ebi.ac.uk/Tools/pfa/phobius/). The

conserved GFFxG motif is indicated. Arrows indicate *H. pylori* Glu-71 and Glu-72. Numbers indicate the lengths of the FlgV homologs.
(TIF)

**S4 Fig. Electron micrographs and motility of *H. pylori* G27M strains used in the study.** (**A**) Transmission electron micrographs of representative flagellated cells of *H. pylori* G27M; *H. pylori* G27M Δ*flgV* mutant; and *H. pylori* G27M Δ*flgV* mutant bearing the pHel3 vector that expresses wild-type FlgV, FlgV$^{F15A/F16A}$, or FlgV$^{E71A/E72A}$. (**B**) Motility of the *H. pylori* G27M derived strains in soft agar medium. Photograph of motility plate was taken 7 d post-inoculation.
(TIF)

**S5 Fig. Alignment of amino acid sequences of predicted C-terminal transmembrane helices of FliF homologs.** FliF homologs are from *Arcobacter butzleri* 7h1h (Abu), *Hydrogenimonas thermophila* (Hth), *Sulfurimonas autotrophica* DSM 16294 (Sau), *Sulfuricurvum kujiense* DSM 16994 (Sku), *Campylobacter jejuni* 81–176 (Cje), *Sulfurospirillum multivorans* DSM 12446 (Smu), *Helicobacter pylori* G27 (Hpy), *Thiovulum* sp. ES (Tsp), *Wolinella succinogenes* DSM 1740 (Wsu), and *Helicobacter hepaticus* ATCC 51449 (Hhe). Alignment was done using Clustal Omega (https://www.ebi.ac.uk/Tools/msa/clustalo/). Transmembrane helices were predicted using Phobius (https://www.ebi.ac.uk/Tools/pfa/phobius/). Arrows indicate *H. pylori* FliF Phe-466 and Tyr-467. Numbers indicate the length of the predicted transmembrane helices.
(TIF)

**S6 Fig. Motility of *H. pylori* G27M expressing FlgV-YFP.** Motilities of *H. pylori* G27M expressing wild-type FlgV (G27M) or FlgV-YFP fusion protein (G27M::*flgV-yfp*) in soft agar medium. Strains were stab inoculated into soft agar medium, and diameters of the resulting swim halos were measured following 7-d incubation. Bars indicate mean values for swim halo diameters. Six to eight replicates were done for each strain. Error bars indicate standard deviation of the mean. The mean swim halo diameters of the two strains did not differ significantly from each other (*p*-value = 0.37). Statistical analysis of the data was done using a two-sample *t* test.
(TIF)

**S1 Table. Primers used in this study.**
(DOCX)

**S2 Table. Intragenic mutations identified in *H. pylori* G27M.**
(DOCX)

**S3 Table. Intergenic mutations identified in *H. pylori* G27M.**
(DOCX)

## Acknowledgments

We thank Jennifer Aronson for critical reading of the manuscript.

## Author Contributions

**Conceptualization:** Timothy R. Hoover.

**Formal analysis:** Jack M. Botting, Shoichi Tachiyama, Katherine H. Gibson, Jun Liu.

**Funding acquisition:** Jun Liu, Timothy R. Hoover.

**Investigation:** Jack M. Botting, Shoichi Tachiyama, Katherine H. Gibson.

**Supervision:** Jun Liu, Vincent J. Starai, Timothy R. Hoover.

**Writing – original draft:** Jack M. Botting.

**Writing – review & editing:** Jack M. Botting, Katherine H. Gibson, Jun Liu, Timothy R. Hoover.

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
