## [Decision Letter · Decision Letter 0]

17 Apr 2023

PONE-D-23-09244Novel FlgV ring is required for optimal flagellar motility of Helicobacter pyloriPLOS ONE

Dear Dr. Hoover,

Thank you for submitting your manuscript to PLOS ONE. After careful consideration, we feel that it has merit but does not fully meet PLOS ONE’s publication criteria as it currently stands. Therefore, we invite you to submit a revised version of the manuscript that addresses the points raised during the review process.  Please submit your revised manuscript by Jun 01 2023 11:59PM. If you will need more time than this to complete your revisions, please reply to this message or contact the journal office at plosone@plos.org.Please include the following items when submitting your revised manuscript:A rebuttal letter that responds to each point raised by the academic editor and reviewer(s). You should upload this letter as a separate file labeled 'Response to Reviewers'.A marked-up copy of your manuscript that highlights changes made to the original version. You should upload this as a separate file labeled 'Revised Manuscript with Track Changes'.An unmarked version of your revised paper without tracked changes. You should upload this as a separate file labeled 'Manuscript'.If applicable, we recommend that you deposit your laboratory protocols in protocols.io to enhance the reproducibility of your results. Protocols.io assigns your protocol its own identifier (DOI) so that it can be cited independently in the future. For instructions see: https://journals.plos.org/plosone/s/submission-guidelines#loc-laboratory-protocols. Additionally, PLOS ONE offers an option for publishing peer-reviewed Lab Protocol articles, which describe protocols hosted on protocols.io. Read more information on sharing protocols at https://plos.org/protocols?utm_medium=editorial-email&utm_source=authorletters&utm_campaign=protocols.

We look forward to receiving your revised manuscript.

Kind regards,

Eric Cascales

Academic Editor

PLOS ONE

Journal Requirements:

  "We thank Jennifer Aronson for critical reading of the manuscript. This work was supported by NIH grants AI140444 and AI146907 to T.R.H. and AI087946 and AI132818 to J.L"

  "This work was supported by National Institutes of Health grants AI140444 (TRH), AI1469077 (TRH), AI087946 (JL), and AI132828 (JI). The URL for NIH is https://www.nih.gov/. The funders had no role in study design, data collection and analysis, decision to publish, or preparation of the manuscript."

4. Please include a copy of Table 4 which you refer to in your text on page 18.

Additional Editor Comments:

It has been sent to two external reviewers. As you will see in their comments pasted below, the two reviewers have a very positive appraisal of your work and recommend publication pending minor revisions. I encourage you to carefully modify your manuscript in light of the reviewer's comments. 

Reviewers' comments:

Reviewer's Responses to Questions

**Comments to the Author**

1. Is the manuscript technically sound, and do the data support the conclusions?

Reviewer #1: Yes

Reviewer #2: Yes

2. Has the statistical analysis been performed appropriately and rigorously? 

Reviewer #1: N/A

Reviewer #2: Yes

3. Have the authors made all data underlying the findings in their manuscript fully available?

Reviewer #1: Yes

Reviewer #2: No

4. Is the manuscript presented in an intelligible fashion and written in standard English?

Reviewer #1: Yes

Reviewer #2: Yes

5. Review Comments to the Author

Reviewer #1: This manuscript describes the identification of protein that forms a previously uncharacterised ring structure on the flagellar motor of Helicobacter pylori of the Campylobacterota. The authors found that FlgV is required for full flagellation and motility in H. pylori, and proceeded to demonstrate using cryoelectron tomography that FlgV forms a cytoplasmic ring. The paper is overall reasonably well-written and essentially scientifically sound, and presents a nice advance in putting the pieces of the puzzle together that at the structures of the Campylobacterota flagella motors.

- The title is grammatically poorly formed, and requires an article ("A novel ring of FlgV"?) - although "novel" is over-used and essentially meaningless. How about "A cytoplasmic ring of FlgV is required for optimal..."?

- The abstract is misleading, hinting that this is the first time FlgV has been shown to be required for motility: "Here, we provide the first evidence that deletion of flgV in H. pylori B128 and a highly motile variant of H. pylori G27 (G27M) results in reduced motility in soft agar medium.". This fails to capture the fact that FlgV has already been characterised. Gao 2014 identified FlgV using a screen that essentially selected for less motile variants, and as they stated: "Although some mutations could indirectly lead to lack of motility, through a variety of functional, biochemical, and in vivo imaging analyses we have provided strong evidence that the genes we have identified encode factors directly involved in the assembly and/or function of the flagellar apparatus."

- Line 44: MotA has four transmembrane helices, not domains.

- Line 48: It is speculation than MotB rotates within MotA - don't present as fact.

- Line 50: "Campylobacterota", not "H. pylori"; worth citing https://mbio.asm.org/content/11/1/e02286-19

- Line 91: worth reporting identity/similarity percentages here.

- Line 99: "strongly suggest" is more appropriate than "strong evidence".

- Line 128: Worth highlighting to non-expert readers that Arcobacter has a mystery, "weird" outlier motor, so this result not expected, but this protein is likely to be FlgV.

- Line 131: I think G27M is a new result? Phrase it as such by writing this sentence in present tense (or provide citation if it's a previous result).

- Entire paragraph starting Line 131: I don't understand why you did this work in non-standard strain G27M instead of G27? Was it because you can more sensitively assay motility using G27M? Spell it out to your readers. I'd have preferred you to have mentioned the WT B128 result first, THEN move into a non-standard strain, although don't understand why you used B128 instead of the correct comparison, G27?

- Line 185: Glu 71,72 come out of the blue - need introduction.

- Paragraph commencing line 196: this is currently fantasy/speculation. I'd prefer the authors either support this hypothesis with relevant fliF mutant results, or remove or greatly tone-down this paragraph. Use words like "suggestive" and "speculative" to help guide your readers to understand the underlying science.

- Line 221: worth acknowledging that this density was previously highlighted in Campylobacter jejuni by the yellow arrows in Fig. 2 of http://www.pnas.org/content/early/2016/03/09/1518952113 and that your work is nicely consistent/explanatory with previous work.

- Line 237: better phrased, "To further assess whether FlgV forms..." (in science we don't try to obtain evidence for a specific hypothesis, as this biases results - rather, we put hypotheses to the test).

- Figure 4: labels need to be larger.

Reviewer #2: This paper aimed to characterise the role of the protein FlgV, a component of the bacterial flagellum, in the human pathogen H. pylori. This protein had previously been identified in C. Jejuni, but its role if flagellum biogenesis and/or function had not been determined. Here, the authors demonstrate that FlgV plays an important role in motility, by promoting flagellum assembly, but also contributes to the flagellum rotation mechanism. Modelling of this protein suggests that it is likely an integral membrane protein, that interacts with the the flagellum MS-ring protein FliF. Finally, very elegant sub-tomogram averaging experiments link FlgV to unattributed density between the MS-ring, and the C-ring, in the cytoplasm.

Collectively, this is a short study, that fills a important gap in our understanding of high-torque-generating flagella such as those of C. Jejune and H. pylori. The manuscript is clear and well-written, and I only have a few minor comments, that would help with clarity and completeness of the reported work.

- Lines 64-66: Some of the mentioned symmetries were shown to be artefacts of FliF in isolation, and were not not found in FliF within the full flagellum HBB structure (Tan et al, Cell, 2021). The true oligomeric state of FliF is 34, with the inner ring (RBM2) having 23-fold symmetry.

- Figure 1b: What does the last operon correspond to?

- Lines 96-97: I think it would be relevant to show the predicted TM topology, in the supplementary material.

- Figure 2C: For AlphaFold modelling, it's essential to show the pLDDT plot, and/or the model Color-coded by confidence, to get a sense of how accurate the prediction is.

- Figure 2: For completeness, a photo of agar plate, a selection of electron micrograph of cells used for counting the flagella, and a table including the number of flagella for each mutant, should be added in the supplementary material.

- Lines 179-183: Showing a multiple alignment of the FlgV sequences, with the GxxxG motif indicated and the predicted tomopogy, would be really helpful.

- Figure 3I, J: These would gain to have additional labels for the FliL ring, which is only mentioned in the legend.

- Figure 3: Could the authors do a map subtraction, between WT and DflgV, and also between WT and flgV-YFP, to better highlight the regions of density that have changed?

- Line 250: This could gain from additional labels, an/or a self-rotation analysis to better illustrate the proposed stoichiometry.

- Figure 4: The model suggests that FlgV is in partly in the IM - should be shown in 4G. Does the non-TM region of the protein fits in the density?

6. PLOS authors have the option to publish the peer review history of their article (what does this mean?). If published, this will include your full peer review and any attached files.

Reviewer #1: No

Reviewer #2: **Yes: **Julien Bergeron

---

## [Author Response · Author response to Decision Letter 0]

10 May 2023

Dear Dr. Cascales,

Thank you for handling the review of the manuscript. We have modified the manuscript based on your comments and the comments of the two reviewers. Below is a point-by-point description of how we addressed the comments, starting with your comments. Our responses follow each specific comment and are in red. We thank both you and the reviewers for the constructive comments as we feel that the suggested changes have significantly improved the manuscript.

Sincerely,

Tim Hoover

Journal Requirements: 

We have reformatted the manuscript to meet the PLOS ONE style requirements.

 "We thank Jennifer Aronson for critical reading of the manuscript. This work was supported by NIH grants AI140444 and AI146907 to T.R.H. and AI087946 and AI132818 to J.L"

 "This work was supported by National Institutes of Health grants AI140444 (TRH), AI1469077 (TRH), AI087946 (JL), and AI132828 (JI). The URL for NIH is https://www.nih.gov/. The funders had no role in study design, data collection and analysis, decision to publish, or preparation of the manuscript."

The funding statement has been removed from the Acknowledgements Section. The Funding Statement indicated above is correct and does not require any amendment.

We removed the information that was referred to by “data not shown” – see lines 233 and 370 on Revised Manuscript with Track Changes. We feel that deleting these statements has no impact on the data presented in the manuscript or interpretation of the results of the study.

4. Please include a copy of Table 4 which you refer to in your text on page 18.

“Table 4” was a typographical error in the original manuscript. This is Table 1.

The information has been added.

The references have been checked.

Reviewer #1: 

- The title is grammatically poorly formed, and requires an article ("A novel ring of FlgV"?) - although "novel" is over-used and essentially meaningless. How about "A cytoplasmic ring of FlgV is required for optimal..."?

The title has been changed to “FlgV forms a flagellar motor ring that is required for optimal motility of Helicobacter pylori”. We have also removed “novel” throughout the manuscript.

- The abstract is misleading, hinting that this is the first time FlgV has been shown to be required for motility: "Here, we provide the first evidence that deletion of flgV in H. pylori B128 and a highly motile variant of H. pylori G27 (G27M) results in reduced motility in soft agar medium.". This fails to capture the fact that FlgV has already been characterised. Gao 2014 identified FlgV using a screen that essentially selected for less motile variants, and as they stated: "Although some mutations could indirectly lead to lack of motility, through a variety of functional, biochemical, and in vivo imaging analyses we have provided strong evidence that the genes we have identified encode factors directly involved in the assembly and/or function of the flagellar apparatus."

We changed the wording of the indicated sentence by replacing “provide the first evidence” to “confirm”. (line 26 in copy with track changes)

- Line 44: MotA has four transmembrane helices, not domains.

The change was made (line 44 of copy with track changes).

- Line 48: It is speculation than MotB rotates within MotA - don't present as fact.

We are not completely clear about the reviewer’s objection, but we modified the sentence to indicate that the MotA pentamer is thought to rotate around the stationary Mot dimer (line 49 of copy with track changes).

- Line 50: "Campylobacterota", not "H. pylori"; worth citing https://mbio.asm.org/content/11/1/e02286-19

We made the change and included the citation (line 51 of copy with track changes).

- Line 91: worth reporting identity/similarity percentages here.

We included the information on identity/similarity (line 226 of copy with track changes).

- Line 99: "strongly suggest" is more appropriate than "strong evidence".

The change was made (line 235 of copy with track changes).

- Line 128: Worth highlighting to non-expert readers that Arcobacter has a mystery, "weird" outlier motor, so this result not expected, but this protein is likely to be FlgV.

We included an additional sentence to indicate that the Arcobacter flagellar proteins are divergent to flagellar proteins of other Campylobacterota and included a reference (lines 263-265 of copy with track changes).

- Line 131: I think G27M is a new result? Phrase it as such by writing this sentence in present tense (or provide citation if it's a previous result).

See response to next comment.

- Entire paragraph starting Line 131: I don't understand why you did this work in non-standard strain G27M instead of G27? Was it because you can more sensitively assay motility using G27M? Spell it out to your readers. I'd have preferred you to have mentioned the WT B128 result first, THEN move into a non-standard strain, although don't understand why you used B128 instead of the correct comparison, G27?

We indicate the reason we enriched for a variant of G27 that displayed greater motility in soft agar medium (lines 269-274 of copy with track changes). We do not feel that starting with B128 and then moving to G27M makes as much sense since most of the data presented in the manuscript was obtained with G27M.

- Line 185: Glu 71,72 come out of the blue - need introduction.

We included an introduction for these amino acid residues (line 336 of the copy with track changes).

- Paragraph commencing line 196: this is currently fantasy/speculation. I'd prefer the authors either support this hypothesis with relevant fliF mutant results, or remove or greatly tone-down this paragraph. Use words like "suggestive" and "speculative" to help guide your readers to understand the underlying science.

We disagree that the paragraph needs to be toned down as we feel it is clear that this is a model based on predicted protein-protein interactions. We agree with the reviewer that generating mutations in fliF would be a reasonable way to examine the validity of the model, and we hope to do those experiments in the future. To support the confidence of the model, we included a figure showing the predicted aligned error (PAE) score for the model for interactions between FlgV and FliF (Fig 2C), as per Reviewer #2’s recommendation.

- Line 221: worth acknowledging that this density was previously highlighted in Campylobacter jejuni by the yellow arrows in Fig. 2 of http://www.pnas.org/content/early/2016/03/09/1518952113 and that your work is nicely consistent/explanatory with previous work.

We included the suggested statement (lines 372-373 of the copy with track changes).

- Line 237: better phrased, "To further assess whether FlgV forms..." (in science we don't try to obtain evidence for a specific hypothesis, as this biases results - rather, we put hypotheses to the test).

The sentence was modified (line 392 of the copy with track changes).

- Figure 4: labels need to be larger.

The labels on Figure 4 have been enlarged.

Reviewer #2:

- Lines 64-66: Some of the mentioned symmetries were shown to be artefacts of FliF in isolation, and were not not found in FliF within the full flagellum HBB structure (Tan et al, Cell, 2021). The true oligomeric state of FliF is 34, with the inner ring (RBM2) having 23-fold symmetry.

We made the changes in the Introduction (lines 66-68 of the copy with track changes) and Discussion (lines 471 and 475 of the copy with track changes).

- Figure 1b: What does the last operon correspond to?

The figure has been fixed.

- Lines 96-97: I think it would be relevant to show the predicted TM topology, in the supplementary material.

A figure showing the predicted membrane topology has been included (S1 Fig).

- Figure 2C: For AlphaFold modelling, it's essential to show the pLDDT plot, and/or the model Color-coded by confidence, to get a sense of how accurate the prediction is.

A figure showing the PAE score has been added (Fig 2C). The original Figure 2C is not Figure 2D.

- Figure 2: For completeness, a photo of agar plate, a selection of electron micrograph of cells used for counting the flagella, and a table including the number of flagella for each mutant, should be added in the supplementary material.

A figure showing representative electron micrographs of flagellated cells of G27M and G27M-derived strains, together with a photo of a plate for the soft agar motility assays has been added (S4 Fig).

- Lines 179-183: Showing a multiple alignment of the FlgV sequences, with the GxxxG motif indicated and the predicted tomopogy, would be really helpful.

A figure showing multiple alignments of FlgV homologs has been included (S3 Fig).

- Figure 3I, J: These would gain to have additional labels for the FliL ring, which is only mentioned in the legend.

The FliL-rings are indicated in the images.

- Figure 3: Could the authors do a map subtraction, between WT and DflgV, and also between WT and flgV-YFP, to better highlight the regions of density that have changed?

We used "map subtraction" between WT and the mutant to better highlight the FlgV ring as shown in the NEW Figure 4E, F. The difference between WT and flgV-YFP could be better visualized in Fig. 3D.

- Line 250: This could gain from additional labels, an/or a self-rotation analysis to better illustrate the proposed stoichiometry.

We included additional labels. The symmetries of the C-ring and the FlgV-ring are 39 and 21, respectively.

- Figure 4: The model suggests that FlgV is in partly in the IM - should be shown in 4G. Does the non-TM region of the protein fits in the density?

FlgV is inserted in the curved inner membrane. We also resolved 21-fold symmetry of the FlgV ring. However, we do not know how many subunits of FlgV are required for the formation of the FlgV ring or if there are other proteins associated with the density that we attribute to FlgV. Therefore, we do not feel convinced to fit the FlgV model into the density.

---

## [Decision Letter · Decision Letter 1]

7 Jun 2023

FlgV forms a flagellar motor ring that is required for optimal motility in Helicobacter pylori

PONE-D-23-09244R1

Dear Dr. Hoover,

Thank you for addressing the reviewer's comments and for the submission of the revised manuscript. It has been sent back to the two original reviewers who noticed that you have appropriately addressed their comments and now recommend publication of your work. I am therefore please to accept your manuscript for publication. Please note that it will be formally accepted once it meets all outstanding technical requirements.

Kind regards,

Eric Cascales

Academic Editor

PLOS ONE

Additional Editor Comments (optional):

Reviewers' comments:

Reviewer's Responses to Questions

**Comments to the Author**

1. If the authors have adequately addressed your comments raised in a previous round of review and you feel that this manuscript is now acceptable for publication, you may indicate that here to bypass the “Comments to the Author” section, enter your conflict of interest statement in the “Confidential to Editor” section, and submit your "Accept" recommendation.

Reviewer #1: All comments have been addressed

Reviewer #2: All comments have been addressed

2. Is the manuscript technically sound, and do the data support the conclusions?

Reviewer #1: Yes

Reviewer #2: Yes

3. Has the statistical analysis been performed appropriately and rigorously? 

Reviewer #1: Yes

Reviewer #2: Yes

4. Have the authors made all data underlying the findings in their manuscript fully available?

Reviewer #1: Yes

Reviewer #2: Yes

5. Is the manuscript presented in an intelligible fashion and written in standard English?

Reviewer #1: Yes

Reviewer #2: Yes

6. Review Comments to the Author

Reviewer #1: I am satisfied with the Author's response to my suggestions and am happy with the manuscript being published.

Reviewer #2: The authors have addressed all the points from both reviewers, and I am happy to recommend the publication of this manuscript in PLoS One.

7. PLOS authors have the option to publish the peer review history of their article (what does this mean?). If published, this will include your full peer review and any attached files.

Reviewer #1: No

Reviewer #2: **Yes: **Julien Bergeron

---

## [Editor Report · Acceptance letter]

5 Jul 2023

PONE-D-23-09244R1 

FlgV forms a flagellar motor ring that is required for optimal motility of *Helicobacter pylori*

Dear Dr. Hoover:

I'm pleased to inform you that your manuscript has been deemed suitable for publication in PLOS ONE. Congratulations! Your manuscript is now with our production department. 

Kind regards, 

on behalf of

Dr. Eric Cascales 

Academic Editor

PLOS ONE